# Sirolimus Pharmacokinetics Variability Points to the Relevance of Therapeutic Drug Monitoring in Pediatric Oncology

**DOI:** 10.3390/pharmaceutics13040470

**Published:** 2021-03-30

**Authors:** Amelia-Naomi Sabo, Sarah Jannier, Guillaume Becker, Jean-Marc Lessinger, Natacha Entz-Werlé, Véronique Kemmel

**Affiliations:** 1Laboratoire de Biochimie et Biologie Moléculaire, Hôpitaux Universitaires de Strasbourg, 67200 Strasbourg, France; amelia-naomi.sabo@chru-strasbourg.fr (A.-N.S.); Jean-Marc.LESSINGER@chru-strasbourg.fr (J.-M.L.); 2Laboratoire de Pharmacologie et Toxicologie Neurocardiovasculaire, Unité de Recherche 7296, Faculté de Médecine de Maïeutique et des Métiers de la Santé, Centre de Recherche en Biomédecine de Strasbourg (CRBS), 67085 Strasbourg, France; guillaume.becker@chru-strasbourg.fr; 3Unité d’Onco-Hématologie Pédiatrique, Hôpitaux Universitaires de Strasbourg, 67200 Strasbourg, France; sarah.jannier@chru-strasbourg.fr; 4Service de la Pharmacie, Hôpitaux Universitaires de Strasbourg, 67200 Strasbourg, France; 5Unité Mixte de Recherche (UMR) 7021, Centre National de la Recherche Scientifique (CNRS), Laboratoire de Bioimagerie et Pathologies, Signalisation Tumorale et Cibles Thérapeutiques, Faculté de Pharmacie, Université de Strasbourg, 67401 Illkirch, France

**Keywords:** sirolimus, pharmacokinetics, therapeutic drug monitoring, pediatric oncology, Monolix^®^ software, pharmacokinetic population modeling

## Abstract

Sirolimus is widely used in transplantation, where its therapeutic drug monitoring (TDM) is well established. Evidence of a crucial role for sirolimus in the PI3K/AkT/mTor pathway has stimulated interest in its involvement in neoplasia, either as monotherapy or in combination with other antineoplastic agents. However, in cancer, there is no consensus on sirolimus TDM. In the RAPIRI phase I trial, the combination sirolimus + irinotecan was evaluated as a new treatment for refractory pediatric cancers. Blood sampling at first sirolimus intake (D1) and at steady state (D8), followed by LC/MS^2^ analysis, was used to develop a population pharmacokinetic model (Monolix^®^ software). A mono-compartmental model with first-order absorption and elimination best fit the data. The only covariate retained for the final model was “body surface area” (D1 and D8). The model also demonstrated that 1.5 mg/m^2^ would be the recommended sirolimus dose for further studies and that steady-state TDM is necessary to adjust the dosing regimen in atypical profiles (36.4% of the population). No correlation was found between sirolimus trough concentrations and efficacy and/or observed toxicities. The study reveals the relevance of sirolimus TDM in pediatric oncology as it is needed in organ transplantation.

## 1. Introduction

Sirolimus, also known as rapamycin, is a macrolide compound widely used for its immunosuppressive action in organ transplantation, alone or in addition to cyclosporine or tacrolimus [1]. It acts by the inhibition of the mammalian serine–threonine kinase target of rapamycin (m-TOR) protein. This action interrupts mitogenic signaling pathways by inactivating cyclins allowing the transition from G1 to S phase and the synthesis of proteins necessary for cell cycle progression. Sirolimus also plays a key role in lipid and glucose metabolism and regulates Adenosine Triphosphate (ATP) production by inhibiting many glycolytic genes, through the inhibition of the transcription factor hypoxia-inducible factor 1α (HIF1α) [2]. Evidence of m-TOR’s crucial role in cellular metabolism and PI3K/AkT/m-TOR signaling pathways stimulated interest in its possible involvement in neoplasia. Preclinical studies have shown the antitumor activity of sirolimus either alone [3,4] or in combination with other antitumor drugs [5,6,7,8,9,10]. In clinical trials, synergistic antiangiogenic and pro-apoptotic effects against pediatric tumors have been demonstrated by combining m-TOR inhibitors with vinblastine [11], cyclophosphamide and topotecan [12], cixutumumab [13,14,15], irinotecan and temozolomide [16,17], or celecoxib, and alternating etoposide and cyclophosphamide [18].

Based on preclinical experiments demonstrating the synergistic effects of sirolimus and irinotecan on the proliferation and metabolism of three tumor cell lines, even at low doses [10], our group tested the combination of daily sirolimus and bimonthly irinotecan administered to pediatric patients (age < 21) with refractory solid tumors in a Phase I trial named RAPIRI (RAPamycin plus IRInotecan) [19]. The aims of the trial were to determine the maximum tolerated dose of this new drug combination, to evaluate the safety and efficacy profiles of each molecule, and at last to determine the doses at which these drugs can be administered.

Sirolimus pharmacokinetics are well known in organ transplantation: its maximum blood concentration occurs 1 to 2 h after oral administration, its bioavailability is low (around 15%), and its effective half-life allows a steady-state concentration to be reached at 5 to 7 days. Sirolimus is known to be a substrate for both cytochrome P-450 (CYP3A4) and p-glycoprotein [20,21]. Therefore, hepatic dysfunction and co-administration of inducers/inhibitors of CYP3A4 or p-glycoprotein influence the pharmacokinetics of sirolimus [22,23].

In transplantation, trough concentration is highly correlated with sirolimus exposure, its efficacy, and toxicity [20,21,24]. Therapeutic targets of sirolimus associated with cyclosporine are generally between 5 and 15 µg/L, while trough sirolimus concentrations greater than 15 µg/L are correlated with increased toxicity and less than 5 µg/L with decreased efficacy [21]. However, in cancers, due to the different existing administration regimens and the different types of cancer, there is no clear consensus on therapeutic drug monitoring (TDM) and on the range of trough concentrations to be used.

Herein, we report a population pharmacokinetics study conducted in the RAPIRI phase I trial, which associated sirolimus (once daily) and irinotecan (every 2 weeks) in pediatric refractory or relapsed solid tumors. The primary objective of our study was to investigate the population pharmacokinetics of sirolimus oral solution administrated after the first dose (with irinotecan) and on day eight (at steady state and without irinotecan) of a 28-day cycle and to assess the impact of different factors on the variability of pharmacokinetic parameters. Our secondary aim was to compare the simulated concentration distributions for each dose level of sirolimus based on its ability to achieve the therapeutic range of 5–15 µg/L used in transplantation therapeutic drug monitoring. Our final endpoint was to compare the trough concentrations of sirolimus of the patients with clinical efficacy or toxicities observed during the trial in order to define the optimal sirolimus starting dose and the therapeutic monitoring strategy relevant for future trials and more generally for therapeutic care using sirolimus as a treatment in pediatric oncology.

## 2. Materials and Methods

### 2.1. Study Design: Drug Combinations and Administration

The RAPIRI phase I trial (national program for clinical research PHRC-N Cancerology 2010, HUS n°4791/n°Eudract: 2010-022329-13, NCT01282697) is a multicenter Phase I clinical trial [19]. Patients from 1 to 21 years old with refractory or relapsed solid tumor after conventional therapies were eligible. All patients or their legal guardians signed a written informed consent prior to inclusion, in accordance with the Declaration of Helsinki and standards of Good Clinical Practice.

This phase I trial followed a dose escalation based on a 3 + 3 design with 10 dose levels (Table 1). Sirolimus oral solution of (1 mg/mL) was administered once daily at a dose of 1, 1.5, 2, or 2.5 mg/m^2^. A 90 min intravenous infusion of irinotecan at 125, 200, or 240 mg/m^2^ was administered just prior to sirolimus on the first and fifteenth day of each 28-day cycle.

Patients underwent physical examinations at baseline and every week during the two first courses and every 2 weeks from the third course to the end of the trial. Adverse events were graded from 1 to 4, all along the trial, according to the National Cancer Institute’s Common Terminology Criteria for Adverse Events (NCI-CTCAE, version 3.0) [19]. Disease evaluations were performed at baseline and at the end of the second cycle of treatment. Patients had either a computed tomography (CT) scan or magnetic resonance imaging (MRI), and their tumor responses were based on the Response Evaluation Criteria in Solid Tumors (RECIST 1.1). Two independent radiologists reviewed all imaging studies [19].

### 2.2. Blood Sampling and Analysis

Sirolimus concentrations were measured after the first intake (D1) and at steady state on day eight (D8). On D1, a blood sample was taken before and at 0.5, 1, 2, 3, and 6.5 h after sirolimus oral administration. On D8, seven blood samples were taken before and at 0.5, 1, 1.5, 2, 4, and 8 h after sirolimus oral administration.

Analyses were performed by reversed-phase liquid chromatography using an Agilent 1200 Series LC system (Agilent Technologies, Santa Clara, CA, USA), which consists of a binary pump, a vacuum degasser, an autosampler, a thermostatted column compartment and a solid phase extraction (SPE) on-line of the injected sample (50 µL) followed by introduction into the mass spectrometer (Agilent 6410 triple quadrupole).

The DOSIMMUNE^®^ kit (Alsachim, Illkirch-Graffenstaden, France) was used for the analysis. The kit enables the simultaneous quantification of four immunosuppressant drugs in whole blood (cyclosporin A, sirolimus, tacrolimus, and everolimus). The procedure consisted of deproteinizing a mixture of 25 µL of whole blood and 25 µL of stable labeled internal standards with an extraction reagent. The chromatographic separation was performed with an elution time of 1.3 min. Ammoniated adducts [M^+^NH^4+^] of molecules were used as precursor ions for all analytes. In the positive-ion mode, the monitored multiple reaction transitions (*m*/*z*) were: Sirolimus 931.7 > 864.5; [^13^C, ^2^H_3_]-Sirolimus 935.5 > 864.5.

The method was accredited (Cofrac n°8-3524). The lower limit of quantification was 0.80 µg/L, the upper limit of linearity was 53 µg/L. Intra-assay and inter-assay variations of internal quality controls were inferior to 10%.

### 2.3. Compartmental Pharmacokinetic Analysis

The pharmacokinetic population model and simulations were performed with the nonlinear mixed-effects modeling Monolix^®^ version 2018R1 (Lixoft SAS, Antony, France) based on the stochastic approximation expectation maximization (SAEM) algorithm. The individual parameters were estimated by the Hastings–Metropolis algorithm. The areas under the curve (AUCs) were secondarily estimated for each patient using the Formula (1).
(1)$${\mathrm{AUC}}_{0-\infty }=\frac{\mathrm{Sirolimus}\;\mathrm{amount}}{\mathrm{Individual}\;\mathrm{clearance}}$$

#### 2.3.1. Base Model

For the base model, one- and two-compartment models with first-order absorption and elimination were tested, as well as with saturable elimination processes. Different residual error models were tested (additive, proportional, or combined models). The most appropriate model was select based on the minimum objective function value (OFV), which was the likelihood ratio test in our study. Exponential models described the inter-individual variability (IIV) for the pharmacokinetic parameters.

#### 2.3.2. Covariate’s Selection

A preliminary graphical assessment of the effect of covariates on pharmacokinetic estimates was made. When a relationship emerged, the influence of covariates on the IIV and accuracy parameters was tested. To be retained, a significant variation in OFV was required. A decrease of at least 3.84 (*p* < 0.05, χ^2^, 1 degree of freedom) on forward selection and an increase of at least 7.88 (*p* < 0.005, χ^2^, 1 degree of freedom) on backward deletion were chosen as criteria for covariates’ selection in order to limit the risk of erroneous conclusions about covariate selection [25]. The inclusion of covariates was guided by clinical plausibility.

The continuous covariates were centered on the median and were age, body weight (BW), height, body surface area (BS), body mass index (BMI), and tumor age. The categorical covariates tested were irinotecan dose on day one, sirolimus dose, cotrimoxazole comedication, glucocorticoids comedication, gender, central nervous system (CNS) tumor diagnostic, and age ≥ 12.

#### 2.3.3. Model Validation

Model evaluation was performed using standard graphs of the correlation between predictions and empirical observations (goodness of fit or GOF), the distribution of weighted residuals (WRES) as a function of time and concentration, and the precision of the pharmacokinetic estimates (residual standard error or RSE). The final model was internally validated based on graphical methods such as distribution of the normalized prediction distribution error (NPDE) and visual predictive checks (VPCs). The VPC graph shows a comparison of the median and the 95% prediction interval for predicted data and the corresponding percentiles for observed data over time.

#### 2.3.4. Simulation of Successive Concentrations

Based on multiple simulations (*n* = 1000) of all the individuals in the dataset, the theoretical distribution of the predictions was set for different doses of sirolimus (1, 1.5, 2, or 2.5 mg/m^2^).

### 2.4. Statistical Analyses

Statistical analyses were performed using GraphPad Prism 9^®^ (GraphPad Software, San Diego, CA, USA). Comparisons of different parameters were made using Student’s tests and a *p*-value of less than 0.05 was required to consider statistical significance.

## 3. Results

### 3.1. Population Characteristics

Demographic characteristics and main information on sirolimus administration and comedications are summarized in Table 2. If 42 patients were included in the RAPIRI trial, due to outliers or missing data, only 27 and 34 pharmacokinetic profiles were included in the population analysis on D1 and D8, respectively. All the patients received sirolimus oral solution (1 mg/mL) once daily and an infusion of irinotecan on D1. Eleven patients were on cotrimoxazole on D1 and 15 on D8. Three patients were on glucocorticoids on D1 and D8.

### 3.2. Pharmacokinetic Results

A total of 27 and 34 pharmacokinetics profiles was analyzed on D1 and D8, respectively. There was a high IIV in the collected data, especially on D1. One- and two-compartment models with linear or saturable elimination processes were tested. A one-compartment model with first-order absorption and elimination best fit the data. The two-compartment model improved the fit of the data, but significantly decreased the accuracy of the parameters (RSE > 80%) and was ultimately not retained. Based on the lowest OFV, the selected residual error model and the distribution of parameters were proportional and log-normal, respectively. Figure 1 shows the goodness of fit graphs. The population and individual predictions are roughly distributed around the y = x line. The selection of covariates was made on the basis of the pharmacokinetic model.

Graphical relationships emerged between volume of distribution (V_d_/F) and allometric covariates such as age, height, BS, and BW on D1 and D8. Thus, the median-normalized BS significantly decreased OFV (ΔOFV = 8.31 on D1 and ΔOFV = 24.81 on D8) and was the only covariate retained on V_d_/F (*p* = 0.0143 and *p* = 1.69 × 10^−5^ on D1 and D8, respectively). It decreased V_d_/F IIV from 97.6% to 62.5% on D1 and from 69.8% to 30.3% on D8. Between allometric covariates, there was no significant improvement in IIV clearance (Cl/F) on D1. Despite this, on D8, median-normalized BS was selected as continuous covariate on Cl/F (*p* = 1.03 × 10^−4^). Once BS was selected as a covariate, no other continuous covariate sufficiently decreased the OFV, on either D1 or D8.

Categorical covariates such as patient gender, CNS tumor diagnostic, or age ≥ 12 did not show any effect on pharmacokinetic parameters and neither did sirolimus and irinotecan doses. Cotrimoxazole comedication showed an effect on Cl/F on D1 (*p* = 0.0207) and D8 (*p* = 0.0079), suggesting that patients on cotrimoxazole had reduced sirolimus clearance (Figure 2). Cotrimoxazole comedication also reduced the clearance IIV by 20.9% and 3.6% on D1 and D8, respectively but did not sufficiently reduce OFV (ΔOFV = 7.13 and ΔOFV = 6.05 on D1 and D8, respectively). Therefore, secondary calculated AUCs were not statistically different on either D1 (*p* = 0.7394), or D8 (*p* = 0.2237) (Figure 2). Indeed, there was a trend of lower doses of sirolimus in patients who were on cotrimoxazole (doses on D1 of 1.8 ± 1.0 mg and on D8 of 2.0 ± 1.0, mean ± SD) than in patients without cotrimoxazole prophylaxis (doses on D1 of 2.5 ± 0.6 mg and on D8 of 2.6 ± 0.6, mean ± SD), *p* = 0.054 on D1 and *p* = 0.082 on D8. This doses discrepancy could explain the difference of significance observed for the clearance but not the AUC between groups with or without cotrimoxazole.

The estimation of the base and final models on D1 and D8, the IIV, the residual errors, and the OFV are summarize in Table 3. The selection of covariates improved the accuracy of the parameters and reduced their variabilities on D1 and D8.

The final formulas on D8 for individual parameters Cl/F_i_ and V_d_/F_i_ are given in Equations (2) and (3), respectively, including individual BS (BS_i_), median population BS ($\bar{\mathrm{BS}}$), population parameters (θ_Cl/F_ and θ_Vd/F_), and estimated influential factors of the BS (β__Cl_BS_ and β__Vd_BS_).
(2)$$\mathrm{Cl}/F_{i}=\theta_{\mathrm{Cl}/F}\times {\left(\frac{\mathrm{BSi}}{\bar{\mathrm{BS}}}\right)}^{\beta_{\_\mathrm{Cl}\_\mathrm{BS}}}$$
(3)$$V_{d}/F_{i}=\theta_{\mathrm{Vd}/F}\times {\left(\frac{\mathrm{BSi}}{\bar{\mathrm{BS}}}\right)}^{\beta_{\_\mathrm{Vd}\_\mathrm{BS}}}$$

The graphs of NPDE vs. time and vs. predicted concentrations showed no trend and were uniformly distributed around the y = 0 line as represented in Appendix A. High variability in sampling time was observed on D1 (Appendix A) but not on D8 (Appendix A). VPC plots for sirolimus data on D1 and D8 were used to assess the predictive property of the final model (Figure 3). They showed that the observed concentrations were within in the intervals of predictions. There were some deviations of the observations from the predicted data represented as red circles in Figure 3. These deviations represented 2.5% of the total of the observations and given the relatively small data set and the high variability of the pharmacokinetic profiles, especially on D1, models were considered as acceptable.

### 3.3. Simulated Concentrations and Therapeutic Range

Finally, simulated sirolimus concentrations after four different doses (1.0, 1.5, 2.0, and 2.5 mg/m^2^) taken once daily for eight days were performed. Given the therapeutic range of trough concentrations of sirolimus at steady state (5–15 µg/L), a search was conducted for the dose of sirolimus that best matched the therapeutic range. The corresponding sirolimus simulations and empirical concentrations on D8 are shown in Figure 4a. While the 1.0 mg/m^2^ dose led to sub-therapeutic trough concentrations of sirolimus, the 2.0 and 2.5 mg/m^2^ doses exposed patients to widely variable concentrations (Figure 4c,d), respectively. The 1.5 mg/m^2^ dose induced adequate steady-state exposure and less variability in sirolimus concentrations (Figure 4b). A percentage of 63.6% of patients receiving this dose had a trough concentration in the therapeutic range on D8, while 36.4% were below the lower limit of the therapeutic range. None of the patients receiving 1.0 mg/m^2^ reached the 5–15 µg/L range on D8. Of the patients receiving 2.0 and 2.5 mg/m^2^ of sirolimus per day, 58.3% and 71.4% were within the recommended trough concentration range, respectively. Nevertheless, these dosing regimens induced highly variable pharmacokinetic profiles and overexposed, respectively, 23% and 14% of patients to high sirolimus concentrations for hours, favoring the development of toxicities.

### 3.4. Efficacy and Toxicities’ Assessment after Two Cycles of Treatment

No correlation between tumor responses and sirolimus trough concentrations was observed, despite the tendency both for patients with a partial response to have higher sirolimus blood levels and the fact that approximately 67% of patients with progressive disease have sirolimus concentrations below 5 µg/L.

The most frequently observed adverse events were vomiting, diarrhea, nausea, oral mucositis, and abdominal pain. Only the maximum grade of toxicity observed was plotted in Figure 5. Nine patients presented grade 3 or 4 toxicities that resolved before the start of cycle 2, and none of these events could be linked to sirolimus-specific toxicities. No significant association was established between sirolimus dose or trough concentrations and toxicity on D8, except for one patient (the red outlier in Figure 5) with a very high concentration of sirolimus on D8 who presented grade 4 adverse events including hypokalemia, hypophosphatemia, hypoalbuminemia, diarrhea, nausea, and abdominal pain.

## 4. Discussion

In this study, we established a population pharmacokinetic model of oral sirolimus for pediatric patients with refractory or relapsed solid tumor included in the RAPIRI phase I trial. We defined covariates to reduce OFV, IIV, and RSE of the base model. Moreover, we compare the simulated concentration distributions for each dose level of sirolimus and the sirolimus trough concentrations observed in trial with clinical efficacy and toxicities in order to define the optimal starting dose of sirolimus and to evaluate the relevance of therapeutic drug monitoring strategy for a future phase II trial.

In our study, the pharmacokinetics of sirolimus were analyzed using a compartmental pharmacokinetic approach after first intake (D1) and at steady state (D8) in a pediatric population receiving a daily dose of 1.0, 1.5, 2.0, or 2.5 mg/m^2^ and an infusion of irinotecan at 125, 200, or 240 mg/m^2^ on D1. A mono-compartmental model with first-order absorption and elimination with a proportional residual error model was the best fit for our data. These results confirm previous sirolimus pharmacokinetic studies in pediatric patients with vascular anomalies [26]. Sirolimus pharmacokinetics were also shown to follow a bi-compartmental model in children with recurrent solid tumors [27], but unlike our study, this work was based on rich sampling data on the distribution and the elimination phases [26,28]. As the bioavailability of sirolimus was not assessed in our study, clearance and volume of distribution estimates represent the apparent pharmacokinetic parameters, Cl/F and V_d_/F, respectively. On D1, sirolimus was rapidly eliminated (Cl/F = 23.9 L/h), and V_d_/F was 88.9 L. On D8, the estimate of sirolimus Cl/F (Cl/F = 11.9 L/h for a BS of 1.3 m^2^) was consistent with previous studies at steady state in pediatric patients with neurofibromatosis type 1 (11.8 L/h) [29] and complicated vascular anomalies (Cl/F = 18.5 L/h) [26], but higher than in pediatric renal transplant recipients receiving calcineurin inhibitor co-therapy (Cl/F = 4.8 L/h/m^2^) [30]. High variability in pharmacokinetic parameters was identified in the base model on D1 (IIV = 95.6% (RSE = 19.5%) and IIV = 97.6% (RSE = 24.6%) in Cl/F and V_d_/F, respectively) and on D8 (IIV = 71.8% (RSE = 13.4%) and IIV = 69.8% (RSE = 16.0%) in Cl/F and V_d_/F, respectively). The final evaluation of the model was conducted using internal tools. The GOF plots showed a rough distribution of observations and predictions around the identity line, with a few patients with atypical profiles and erroneous model predictions. The mean and variance of the NPDE were consistent with a normal distribution (*N*(0; 1)). Therefore, a high variability in sampling time was observed on day one, which may be due to the fact that children and nurses were not used to restrictive sampling schedules on the first day of the study. Despite highly variable population profiles, the VPC plots illustrated acceptable model prediction capabilities.

In order to try to explain parameters’ IIV, the influence of covariates on the pharmacokinetic parameters was studied. On D1, even though allometric covariates such as patient age, height, BS, and BW graphically correlated with V_d_/F, the BS was the only one that sufficiently improved the OFV (ΔOFV = 8.31) and was retained for the final model. Its selection explained one-third of the V_d_/F IIV on day one. None of the continuous or categorical covariates could explain the high Cl/F IIV on day one. On day eight, the BS selection on Cl/F and V_d_/F improved the base model (ΔOFV = 41.23) and decreased the parameters IIV (ΔIIV = 21.7% in Cl/F and ΔIIV = 40.4% in V_d_/F). Although age had an effect on clearance on day eight, once BS was retained on the model, the categorical covariate “age ≥ 12” had no effect on the estimate of Cl/F in our population. Goyal et al. showed that weight-normalized Cl/F was three times higher in patients under 12 years of age than in those over 12 years of age in early post myeloablative blood and marrow transplantation, which could be explained by age-dependent variations in enzyme expression and activity [31].

Sirolimus is extensively metabolized by intestinal and hepatic CYP3A4 and is also a substrate of the p-glycoprotein efflux pump [22,23,32]. In literature, administration of strong inhibitors of CYP3A4 such as azoles or diltiazem increases sirolimus trough concentrations [21], as does the co-administration of cyclosporine, a substrate and inhibitor of the p-glycoprotein [33]. In our study, irinotecan, a p-glycoprotein substrate that may compete for p-glycoprotein transport [32,34], was administered during a 90 min infusion in different doses (125, 200, or 240 mg/m^2^) just prior to sirolimus intake on D1. No effect of irinotecan doses was observed on sirolimus pharmacokinetic parameters (*p* > 0.05). Then, three patients were stabilized on glucocorticoid treatments (one patient on dexamethasone and two patients on hydrocortisone), which are inducers of CYP3A4 gene expression [35], but the effect of this categorical covariate could not be correlated with sirolimus pharmacokinetic parameters. However, other studies have shown the same results, suggesting that other etiologies should be explored if sirolimus concentrations change during glucocorticoid treatment [36,37]. Finally, cotrimoxazole is a combination of two antimicrobial agents (sulfamethoxazole and trimethoprim) that act synergistically against a wide variety of bacteria and protozoa and is routinely used as prophylaxis against opportunistic infections. It is known as an in vitro inhibitor of CYP 2C8/2C9 at low concentrations and as an inhibitor of CYP3A4 at high concentrations [38]. Administration of cotrimoxazole prior to sirolimus intake did not influence the maximum whole blood sirolimus concentration or AUC in renal transplant recipients [39]. In our study, 11 patients were on long-term cotrimoxazole on D1, and 15 patients were on D8. The median dose of cotrimoxazole was 400/80 (range from 200/40 to 800/160 mg sulfamethoxazole/trimethoprim on D1 and D8), and cotrimoxazole was always administered three times a week. The time elapsed between the last dose of cotrimoxazole and the measurement of sirolimus concentrations was not noted. Cotrimoxazole was analyzed as a categorical covariate and patients on cotrimoxazole had 2.20-fold (*p* = 0.0207) and 1.71-fold (*p* = 0.0079) lower Cl/F compared to patients who did not take this prophylaxis on D1 and D8, respectively. Therefore, the selection of this covariate was impossible due to an insufficient drop in OFV. Consistent with the previous study [39], the secondary calculated AUCs were not statistically different between the two groups on either D1 (*p* = 0.7394) or D8 (*p* = 0.2237) due to a trend of lower doses of sirolimus in patients on cotrimoxazole. In consequence, despite the decreased clearance of sirolimus, no clinical impact was expected in pediatric patients with refractory solid tumors on long-term cotrimoxazole therapy.

After successive daily administrations, steady-state sirolimus concentrations were reached in 5 to 7 days. Zimmerman and Kahan showed that there is an excellent linear correlation between the trough steady-state blood concentration and AUC (coefficient of determination R^2^ = 0.99, slope = 0.0652, intercept = −0.880), over a dose range of 0.5 to 3.5 mg/m^2^ twice a day, suggesting that the trough blood concentration could be a relevant estimator of the sirolimus exposure in this dose range [40]. Thus, the trough target sirolimus concentrations are between 5 to 15 µg/L, and they are already used in organ transplantation and could certainly be use in oncology. Based on these guidelines, our study investigated the daily dose of sirolimus (1.0, 1.5, 2.0, or 2.5 mg/m^2^) that provides sirolimus steady-state concentrations within the recommended range and showed that the 1.5 mg/m^2^ dose provided the sirolimus concentrations that best met the guidelines with less variability in the concentration profiles. Overall, 63.6% of patients receiving this dose reached the range of 5–15 µg/L on D8. In consistency with our results, other pediatric oncology studies suggested an initial dosing regimen of 1.5 mg/m^2^ per day or 0.8 mg/m^2^ twice a day. However, due to the limited percentage of patients reaching the recommended range and the occurrence of reduced efficacy, side effects, or toxicities, therapeutic drug monitoring is necessary 7 to 10 days after sirolimus initiation, in order to adjust the dose. In consequence, the starting daily dose for a future RAPIRI Phase II trial would be 1.5 mg/m^2^ once a day during a 28-day cycle, and a measurement of the trough steady-state concentration would be necessary on D8 to adjust the sirolimus dose for the outliers of the therapeutic range. Once the adjustment is made, further pre-administration sampling would be required after 1 to 2 weeks until the target range is reached.

## 5. Conclusions

In order to assess the variability of the pharmacokinetic parameters of sirolimus administered daily, a population pharmacokinetic model was proposed after the first dose and at steady state. Explaining some of the IIV, the median-normalized BS was selected as covariate on V_d_/F on day one and on Cl/F and V_d_/F on day eight. Despite the decreased sirolimus clearance in patients receiving long-term cotrimoxazole, the sirolimus AUCs were not modified, and no clinical impact of cotrimoxazole administration was expected in routine. The model data also demonstrated that 1.5 mg/m^2^ would be the recommended dose for a future phase II trial, but therapeutic drug monitoring is necessary at steady state to adjust the sirolimus dosing regimen in atypical patient profiles. Moreover, our study reveals the relevance of sirolimus therapeutic monitoring in pediatric oncology as it is needed in organ transplantation.

## Figures and Tables

**Figure 1 pharmaceutics-13-00470-f001:**
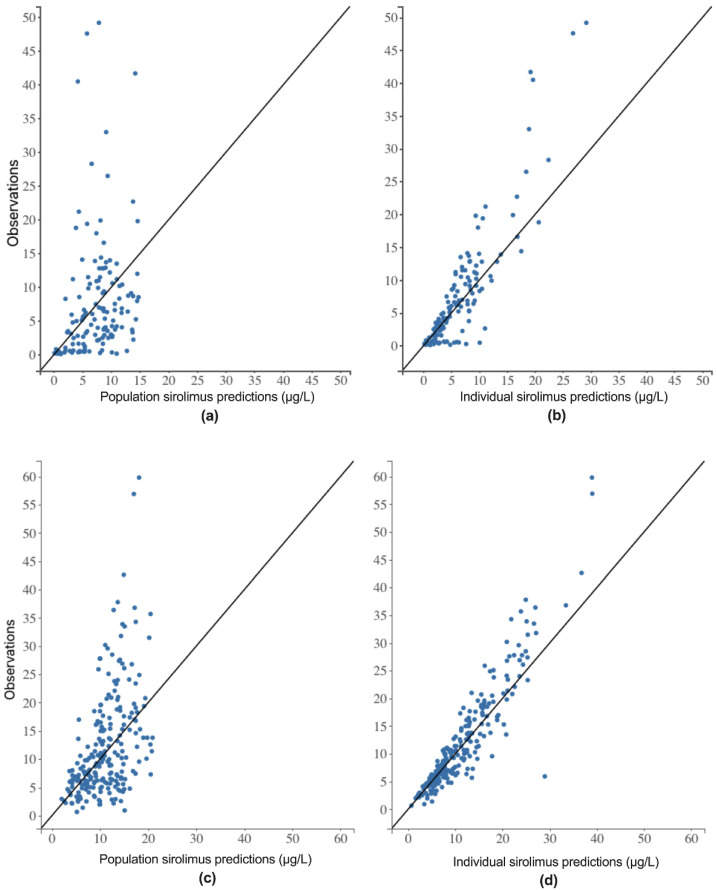
Goodness of fit plots. Observed concentrations versus population predicted concentrations on day 1 (D1) (**a**) and day 8 (D8) (**c**). Observed concentrations versus individual predicted concentrations on D1 (**b**) and D8 (**d**). Black line: y = x.

**Figure 2 pharmaceutics-13-00470-f002:**
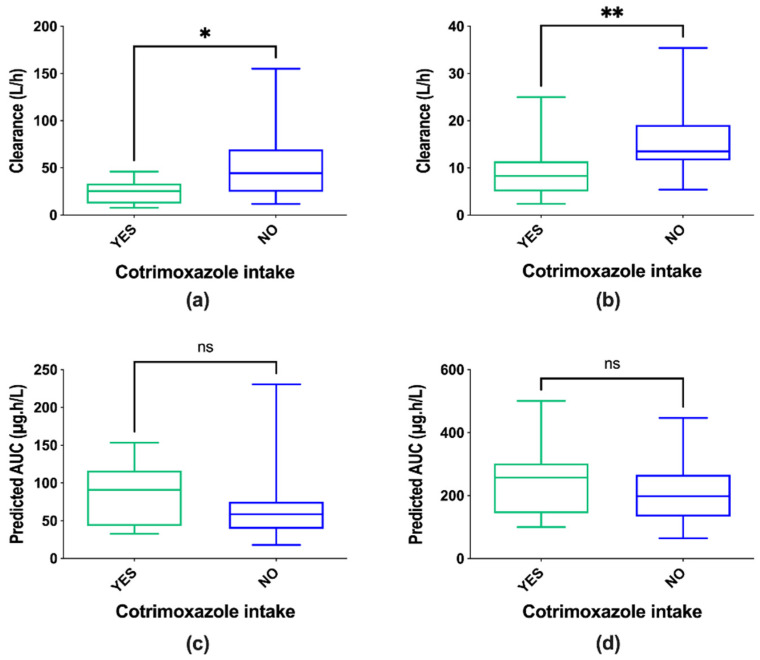
Clearance distribution on D1 (**a**) and D8 (**b**) with or without cotrimoxazole comedication. Exposures to sirolimus (predicted AUCs) were not statistically different on either D1 (**c**) or D8 (**d**) because of a trend of lower doses of sirolimus in patients who were on cotrimoxazole than in patients without cotrimoxazole prophylaxis. Statistical analysis: Unpaired t test, alpha = 0.05. * *p* < 0.05, ** *p* < 0.01, ns = no significant differences.

**Figure 3 pharmaceutics-13-00470-f003:**
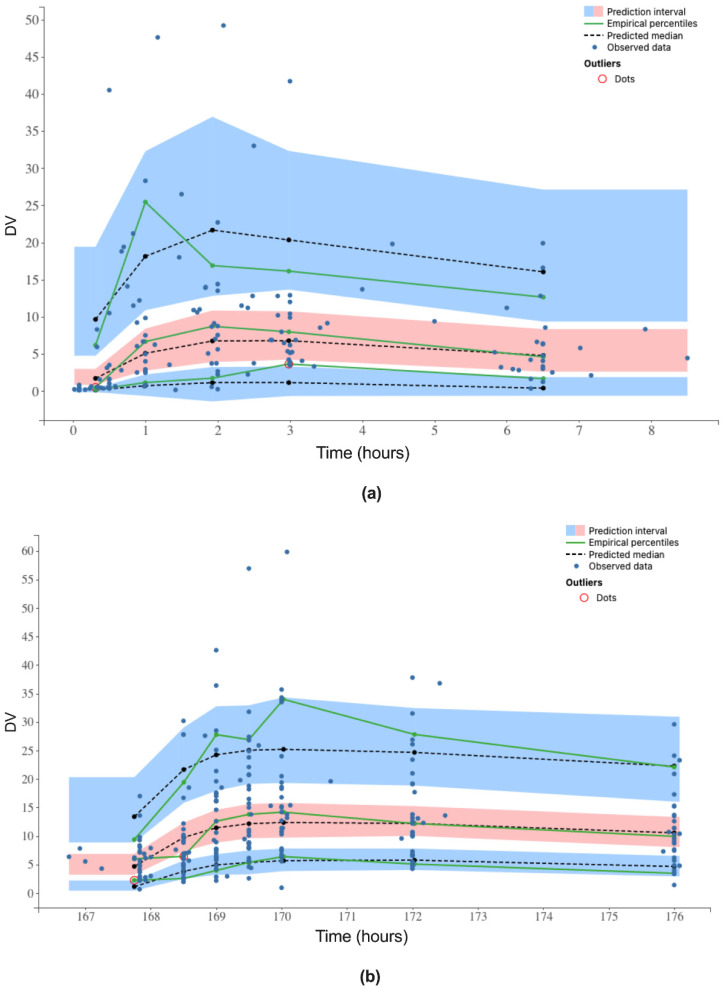
Visual predictive check plots of sirolimus observed concentrations (dependent variable DV in µg/L) versus time after the first sirolimus dose (**a**) and on D8 (**b**). Blue areas are 95% confidence interval of the 10th and 90th percentiles. The pink area is the confidence interval of the median. Black dotted lines represent means of the 10th, 50th, and 90th predicted percentiles. Blue dots represent observations and green continuous lines represent means of the 10th, 50th, and 90th observed percentiles. Red circles show deviations of the predicted data from the observations.

**Figure 4 pharmaceutics-13-00470-f004:**
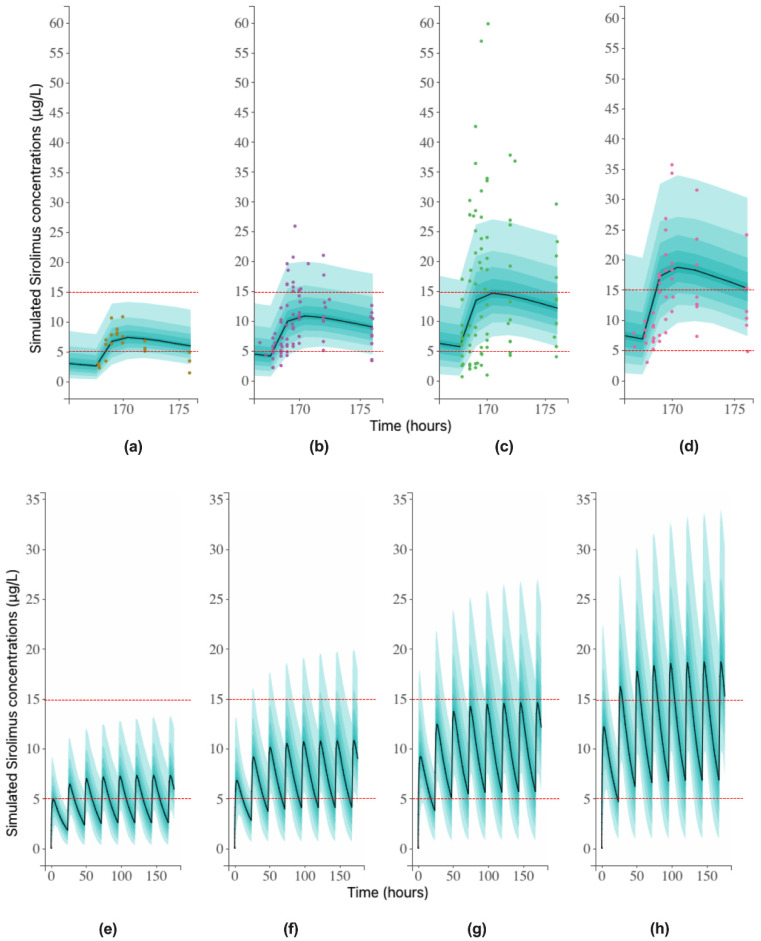
Simulated concentrations versus time after 1.0 mg/m^2^ (**a**), 1.5 mg/m^2^ (**b**), 2.0 mg/m^2^ (**c**), and 2.5 mg/m^2^ (**d**) of sirolimus on D8. Dots represent observed concentrations on D8. Simulated concentrations versus time after 1.0 mg/m^2^ (**e**), 1.5 mg/m^2^ (**f**), 2.0 mg/m^2^ (**g**), and 2.5 mg/m^2^ (**h**) of sirolimus for 8 days. Dashed lines represent the sirolimus therapeutic range (5–15 µg/L). Solid line represents the median, and the nine green bands represent each 10% percentile of the 90% simulated concentrations distribution.

**Figure 5 pharmaceutics-13-00470-f005:**
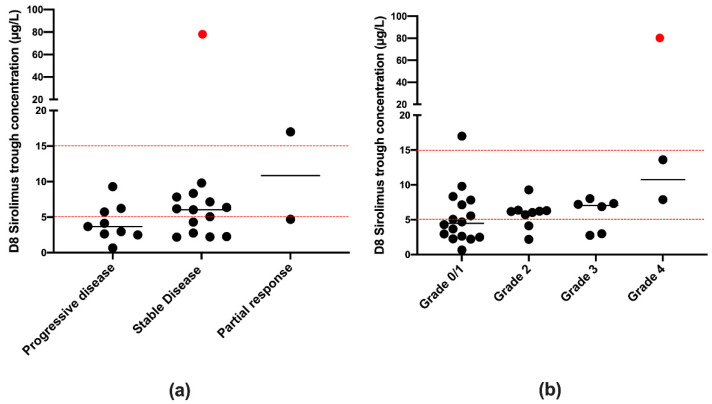
Efficacy and toxicities assessment. Representation of sirolimus trough concentrations at steady state (D8) depending on the disease evolution (**a**) and the toxicity (**b**). The red point is an outlier and corresponds to a very high trough concentration (79.6 µg/L) of a patient presenting stable disease evolution and high toxicity. Red dash lines represent sirolimus therapeutic range used in transplantation (5–15 µg/L).

**Table 1 pharmaceutics-13-00470-t001:** RAPIRI phase I trial characteristic with the levels, doses of irinotecan and sirolimus, and the number of patients included in each level.

Levels	Irinotecan Dose (mg/m^2^, D1)	Sirolimus Dose (mg/m^2^/day)	Included Patients
1	125	1	3
2	125	1.5	6
3	125	2	6
4	125	2.5	3
5	200	1.5	3
6	200	2	3
7	200	2.5	3
8	240	1.5	3
9	240	2	6
10	240	2.5	6

**Table 2 pharmaceutics-13-00470-t002:** Demographic characteristics and main information about sirolimus and comedications.

Variable	D1	D8
**Demographics**		
Number of pharmacokinetic profiles	27	34
Number of observations	127	229
Age (years) (mean ± SD)	11.7 ± 5.9	12.6 ± 5.6
<12 years	14	14
≥12 years	13	20
Male/Female	16/11	21/13
Body weight (kg) (mean ± SD)	38.5 ± 18.6	40.7 ± 17.4
Height (cm) (mean ± SD)	140.9 ± 31.2	146.0 ± 29.25
Body surface area (m^2^) (mean ± SD)	1.2 ± 0.4	1.3 ± 0.4
Body mass index (kg/m^2^) (mean ± SD)	18.1 ± 2.9	18.1 ± 2.8
Central nervous system tumor/other diagnostics	12/15	14/20
**Sirolimus information**		
Real administered dose (mean ± SD, (*n*))		
Dose level group 1 mg/m^2^	1.22 ± 0.81 (2)	1.14 ± 0.59 (3)
Dose level group 1.5 mg/m^2^	2.03 ± 0.67 (10)	2.04 ± 0.63 (11)
Dose level group 2 mg/m^2^	2.26 ± 0.88 (9)	2.54 ± 0.78 (13)
Dose level group 2.5 mg/m^2^	2.78 ± 0.92 (6)	2.93 ± 0.94 (7)
**Irinotecan administration**		
Irinotecan infusion before sirolimus intake	Yes	No
Real administered dose (mean ± SD, (*n*))		
Dose level group 125 mg/m^2^	150 ± 52.3 (14)	-
Dose level group 200 mg/m^2^	307 ± 80.5 (5)	-
Dose level group 240 mg/m^2^	247 ± 102 (8)	-
**Comedication**		
Number of patients on cotrimoxazole Median doses 400/80 mg Range from 200/40 to 800/160 mg	11	15
Number of patients on glucocorticoids	3	3

**Table 3 pharmaceutics-13-00470-t003:** Base and final models estimation on D1 and D8 of sirolimus quantification.

Parameter	Definition	Base Model Estimation (RSE)		Final Model Estimation (RSE)	
Structural model		D1	D8	D1	D8
k_a_ (h^−1^)	Absorption rate constant	0.52 (41.6)	1.02 (27.1)	0.46 (39.2)	0.97 (27.1)
Cl/F (L.h^−1^)	Apparent sirolimus clearance	21.5 (22.8)	10.0 (11.5)	23.9 (21.8)	11.9 (9.2)
V_d_/F (L)	Apparent volume of distribution	92.8 (37.2)	214.0 (15.7)	88.9 (35.5)	238.0 (9.9)
Covariates					
β__Vd_BS_	Effect of body surface on V_d_/F	-	-	1.35 (31.0)	1.41 (15.5)
β__Cl_BS_	Effect of body surface on Cl/F	-	-	-	1.09 (21.2)
IIV (CV)					
IIV-k_a_ (%)	k_a_ interindividual variability	110.4 (26.9)	141.8 (32.2)	80.0 (25.3)	168.5 (21.2)
IIV-Cl/F (%)	Cl/F interindividual variability	95.6 (19.5)	71.8 (13.4)	92.9 (19.2)	50.1 (13.5)
IIV-V_d_/F (%)	V_d_/F interindividual variability	97.6 (24.6)	69.8 (16.0)	62.5 (38.3)	29.4 (25.1)
Residual error					
b (%)	Proportional residual error	0.549 (8.97)	0.312 (6.25)	0.538 (8.90)	0.306 (6.19)
OFV	−2 log-likelihood value	755.97	1361.87	747.66	1320.64

IIV: interindividual variability; CV: coefficient of variation; OFV: objective function value.

## Data Availability

The data presented in this study are available on request from the corresponding author. The data are not publicly available due to ethical restriction because it is young patient data from a clinical trial.

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
