# Peer review of "Sirolimus Pharmacokinetics Variability Points to the Relevance of Therapeutic Drug Monitoring in Pediatric Oncology"

_pharmaceutics, 2021, doi:10.3390/pharmaceutics13040470_

Round 1

Reviewer 1 Report

This clinical study investigated the population pharmacokinetics of Sirolimus oral solution and assessed the impact of different factors on the variability of pharmacokinetic parameters. It also compared the simulated concentration distributions for each dose level of Sirolimus based on their ability to achieve the therapeutic range of 5-15 µg/L used in transplantation therapeutic drug monitoring. The authors found relevance of Sirolimus TDM in drug repositioning for pediatric oncology. This finding is important, as it can help build consensus on therapeutic drug monitoring and on the range of trough concentrations to be used in transplantation.

My only minor suggestion is:

mTor or HIF1 signaling was not specially analyzed in this paper, although sirolimus is known to have inhibitory effects. Please reconsider if the title is accurate.

Author Response

Dear reviewer,

We would like to thank you for your time in evaluating our work and for your constructive comments that helped us to improve the quality of our manuscript.

Our Response :

  • My only minor suggestion is: mTor or HIF1 signaling was not specially analysed in this paper, although sirolimus is known to have inhibitory effects. Please reconsider if the title is accurate.

We change the title in accordance of the article results. And we add some explanation about RAPIRI trial objectives and design in the introduction in term to explain the link between Sirolimus and Irinotecan.

The Title is now : "Sirolimus pharmacokinetics variability points the relevance of therapeutic drug monitoring in pediatric oncology"

Reviewer 2 Report

This manuscript reports results on the accessment of the variability of the pharmacokinetic parameters of sirolimus daily administration and development of population pharmacokinetic model of sirolimus. The PK model was successfully demonstrated and provide the recommended dose of future trial.  However, the extensive rewriting is required to make the readers understand the flow of the manuscript more easily. Also, there are still some suggestions and/or questions to improve the manuscript as below.

  1. In the introduction, there is no information about co-medication design of this studies. In the Table 1, there are three kinds of co-administered drugs such as irinotecan, cotrimoxazole and glucocorticoids. In addition, the reviewer does not find the reason to conduct sirolimus and irinotecan combi-therapy at all in the introduction. There is no sufficient information provided for sirolimus and irinotecan combi-therapy.
  2. Although there are several descriptions about PK model comparison, more discuss are needed compared to those of previous literatures. In terms of pediatric patients, the developed model can be compared particularly.
  3. For the Fig 2, the reviewer hardly understands the discrepancy between CL/F and AUC in the absence and presence of cotrimoxazole. The authors say that cotrimoxazole comedication showed an effect on Cl/F on D1 (p = 0.0207) and D8 (p = 0.0079), suggesting that patients on Cotrimoxazole had reduced Sirolimus clearance. How come the AUCs were not significantly different? This point should be clarified in the revised manuscript.
  4. In the method section of 2.2., the sirolimus analytical LC-MS/MS condition should be provided in detail.
  5. In the Table 1, more information is needed for co-medication, such as dose.
  6. In the abstract, no description about sirolimus plus irinotecan combi-therapy exist, although the title has the words, sirolimus plus irinotecan.
  7. Please check the typos and grammatical errors on the manuscript carefully. Also please check the title as well.

Author Response

Dear Reviewer,

The modifications in the main document were indicated using highlight characters. We would like to thank you for your time in evaluating our work and for your constructive comments that helped us to improve the quality of our manuscript. We work on the quality of the writing to improve its fluency and correct mistakes.

  1. In the introduction, there is no information about co-medication design of this studies. In the Table 1, there are three kinds of co-administered drugs such as irinotecan, cotrimoxazole and glucocorticoids. In addition, the reviewer does not find the reason to conduct sirolimus and irinotecan combi-therapy at all in the introduction. There is no sufficient information provided for sirolimus and irinotecan combi-therapy.

Our Response :

We change the introduction in order to explain the reason of combi-therapy of Sirolimus + Irinotécan. A new Table is also add in order to shown the distribution of doses of Irinotecan and Sirolimus in the population of patients included.

"Table 1. RAPIRI phase I trial characteristic with the levels, doses of Irinotecan and Sirolimus and the number of patients included in each level.

Levels

Irinotecan dose (mg/m², D1)

Sirolimus dose (mg/m²/day)

Included patients

1

125

1

3

2

125

1.5

6

3

125

2

6

4

125

2.5

3

5

200

1.5

3

6

200

2

3

7

200

2.5

3

8

240

1.5

3

9

240

2

6

10

240

2.5

6

The modifications in text are :

Based on preclinical experiments demonstrating synergistic effects of Sirolimus and Irinotecan on the proliferation and metabolism of three tumor cell lines, even at low doses [10], our group tested the combination of daily Sirolimus and bimonthly Irinotecan administered to paediatric patients (age < 21) with refractory solid tumors in a Phase I trial named RAPIRI [19]. The aims of the trial were to determine the maximum tolerated dose  of this new drug combination, to evaluate safety and the efficacy profiles of each molecules and at last to determine the doses at which these drugs can be administered.”

2. Although there are several descriptions about PK model comparison, more discuss are needed compared to those of previous literatures. In terms of pediatric patients, the developed model can be compared particularly.

Our Response :

We change the discussion and add more comparison with pediatric PK studies. We also add some references :

The modifications in text are :

"These results confirm previous Sirolimus pharmacokinetic studies in pediatric patients with vascular anomalies [26]. Sirolimus pharmacokinetics were also shown to follow a bi-compartmental model in children with recurrent solid tumors [26,27], but unlike our study, this work was based on rich sampling data on the distribution and the elimination phases [26,28]. As the bioavailability of Sirolimus was not assessed in our study, clearance and volume of distribution estimates represent the apparent pharmacokinetic parameters, Cl/F and Vd/F, respectively. On D1, Sirolimus was rapidly eliminated (Cl/F = 23.9 L/h) and Vd/F was 88.9 L. On D8, the estimate of Sirolimus Cl/F (Cl/F = 11.9 L/h for a BS of 1.3 m2) was consistent with previous studies at steady-state in pediatric patients with neurofibromatosis type 1 (11.8 L/h) [29] and complicated vascular anomalies (Cl/F = 18.5 L/h) [26], but higher than in pediatric renal transplant recipients receiving calcineurin inhibitor co-therapy (Cl/F = 4.8 L/h/m2) [30].

The references add are :

27. Mizuno, T.; Fukuda, T.; Christians, U.; Perentesis, J.P.; Fouladi, M.; Vinks, A.A. Population Pharmacokinetics of Temsirolimus and Sirolimus in Children with Recurrent Solid Tumours: A Report from the Children’s Oncology Group. British Journal of Clinical Pharmacology 2017, 83, 1097–1107, doi:https://doi.org/10.1111/bcp.13181.

29. Scott, J.R.; Courter, J.D.; Saldaña, S.N.; Widemann, B.C.; Fisher, M.; Weiss, B.; Perentesis, J.; Vinks, A.A. Population Pharmacokinetics of Sirolimus in Pediatric Patients With Neurofibromatosis Type 1. Therapeutic Drug Monitoring 2013, 35, 332–337, doi:10.1097/FTD.0b013e318286dd3f.

30. Schachter, A.D.; Benfield, M.R.; Wyatt, R.J.; Grimm, P.C.; Fennell, R.S.; Herrin, J.T.; Lirenman, D.S.; McDonald, R.A.; Munoz‐Arizpe, R.; Harmon, W.E. Sirolimus Pharmacokinetics in Pediatric Renal Transplant Recipients Receiving Calcineurin Inhibitor Co-Therapy. Pediatric Transplantation 2006, 10, 914–919, doi:https://doi.org/10.1111/j.1399-3046.2006.00541.x.

3. For the Fig 2, the reviewer hardly understands the discrepancy between CL/F and AUC in the absence and presence of cotrimoxazole. The authors say that cotrimoxazole comedication showed an effect on Cl/F on D1 (p = 0.0207) and D8 (p = 0.0079), suggesting that patients on Cotrimoxazole had reduced Sirolimus clearance. How come the AUCs were not significantly different? This point should be clarified in the revised manuscript.

Our Response :

We totally understand the questions of the reviewer and we apologies the lack of clear explanations. It is exact that Cl/F depend of AUC with the equation (1) indicated in the text.

The only variable which can explain the discrepancy between the results obtain with CL/F and AUC was Sirolimus amount and unfortunately there was a difference in Sirolimus dose for patients with and without Cotrimoxazole comedication.

The modifications in the text are :

"Therefore, secondary calculated AUCs were not statistically different on either D1 (p = 0.7394), or D8 (p = 0.2237) (Figure 2). Indeed, there was a trend of lower doses of Sirolimus in patients who were on Cotrimoxazole (doses on D1 of 1.8 ± 1.0 mg and on D8 of 2.0 ± 1.0, mean ± SD) than in patients without Cotrimoxazole prophylaxis (doses on D1 of 2.5 ± 0.6 mg and on D8 of 2.6 ± 0.6, mean ± SD), p = 0.054 on D1 and p = 0.082 on D8.

4. In the method section of 2.2., the sirolimus analytical LC-MS/MS condition should be provided in detail.

Our Response :

We developpe the analytical LC/MS² condition in the text

The modifications are :

"Analyses were performed by reversed-phase liquid chromatography using an Agilent 1200 Series LC system (Agilent Technologies, Santa Clara, California, USA), which consists of a binary pump, vacuum degasser, autosampler and a thermostated column compartment and a SPE on-line of the injected sample (50 µL) followed by introduction into the mass spectrometer (Agilent 6410 triple quadrupole).

The DOSIMMUNE® kit (Alsachim, Illkirch-Graffenstaden, France) was used for the analysis. The kit is enables to simultaneous quantification of 4 immunosuppressant drugs in whole blood (Cyclosporin A, Sirolimus, Tacrolimus and Everolimus). The procedure consisted to deproteinize a mixture of 25 µL of whole blood, 25 µL of stable labeled internal standards with an extraction reagent. The chromatographic separation was performed with an elution time of 1.3 min. Ammoniated adducts [M+NH4+] of molecules were used as precursor ions for all analytes. In the positive-ion mode, the monitored multiple reaction transitions (m/z) were: Sirolimus 931.7 -> 864.5; [13C, 2H3]-Sirolimus 935.5 -> 864.5."

5. In the “old Table 1” or “actual Table 2”, more information is needed for co-medication, such as dose.

Our Response :

We add a Table 1 to explain the 10 levels of co-medication (Sirolimus + Irinotecan) and the number of patients included in each level. And we add the real dose administrated for each Irinotecan doses group in table 2  

Irinotecan administration

Irinotecan infusion before Sirolimus intake

Yes

No

Real administered dose (mean ± SD, (n))

Dose level group 125 mg/m2

150 ± 52.3 (14)

-

Dose level group 200 mg/m2

307 ± 80.5 (5)

-

Dose level group 240 mg/m2

247 ± 102 (8)

-

6.In the abstract, no description about sirolimus plus irinotecan combi-therapy exist, although the title has the words, sirolimus plus irinotecan.

Our Response :

We change the abstract and the title in consequence.

7. Please check the typos and grammatical errors on the manuscript carefully. Also please check the title as well.

We apologies the typos and grammatical errors and we hope, we had correct all those mistakes

Round 2

Reviewer 2 Report

The revised manuscript tries to reflect the reviewer's suggestions and/or questions properly.